# Wasserstein Gradient Flows for Optimizing GMM-based Policies

## Abstract

Robots often rely on a repertoire of previously-learned motion policies for performing tasks of diverse complexities. When facing unseen task conditions or when new task requirements arise, robots must adapt their motion policies accordingly. In this context, policy optimization is the *de facto* paradigm to adapt robot policies as a function of task-specific objectives. Most commonly-used motion policies carry particular structures that are often overlooked in policy optimization algorithms. We instead propose to leverage the structure of probabilistic policies by casting the policy optimization as an optimal transport problem. Specifically, we focus on robot motion policies that build on Gaussian mixture models (GMMs) and formulate the policy optimization as a Wasserstein gradient flow over the GMMs space. This naturally allows us to constrain the policy updates via the $L^2$-Wasserstein distance between GMMs to enhance the stability of the policy optimization process. Furthermore, we leverage the geometry of the Bures-Wasserstein manifold to optimize the Gaussian distributions of the GMM policy via Riemannian optimization. We evaluate our approach on common robotic settings: Reaching motions, collision-avoidance behaviors and multi-goal tasks. Our results show that our method outperforms common policy optimization baselines in terms of task success rate and low-variance solutions.

## 1 Introduction

One of the main premises about autonomous robots is their ability to successfully perform a large range of tasks in unstructured environments. This demands robots to adapt their task models according to environment changes, and consequently to adjust their actions to successfully perform under unseen conditions (Peters et al., 2016). In general, robotic tasks, e.g. picking or inserting an object, are usually executed by composing previously-learned skills (Schaal et al., 2003), each represented by a motion policy. Therefore, in order to successfully perform under new settings, the robot should adapt its motion policies according to the new task requirements and conditions.

Research on methods for robot motion policy adaptation is vast (Kober et al., 2013; Chatzilygeroudis et al., 2020), with approaches mainly building on black-box optimizers (Stulp & Sigaud, 2012), end-to-end deep reinforcement learning (Ibarz et al., 2021), and policy search (Deisenroth et al., 2013). Regardless of the optimization method, most approaches rely on policy structure-unaware adaptation strategies. However, several motion policy models (e.g, dynamic movement primitives (DMP)(Ijspeert et al., 2013), Gaussian mixture models (GMM) (Calinon et al., 2007), probabilistic movement primitives (ProMPs) (Paraschos et al., 2018), and neural networks (Bahl et al., 2020), among others), carry specific physical or probabilistic structures that should not be ignored. First, these policy models are often learned from demonstrations in a starting learning phase (Schaal et al., 2003), thus the policy structure already encapsulates relevant prior information about the skill. Second, structure-unaware adaptation strategies optimize the policy parameters disregarding the special characteristics of the policy model (e.g., a DMP represents a second-order dynamical system). In this regard, we hypothesize that the policy structure may be leveraged to better control the adaptation strategy via policy structure-aware gradients and trust regions.

Our main idea is to design a policy optimization strategy that explicitly builds on a particular policy structure. Specifically, we focus on GMM policies, which have been widely used to learn motion skills from human demonstrations (Calinon et al., 2007; Cederborg et al., 2010; Calinon, 2016;

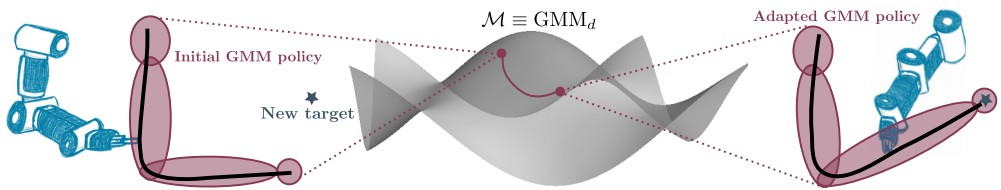

Figure 1: Illustration our policy structure-aware adaptation of GMM policies. Policy updates follow a Wasserstein gradient flow on the manifold of GMM policies $\mathrm{GMM}_d$.

Jaquier et al., 2019). GMMs provide a simple but expressive enough representation for learning a large variety of robot skills: Stable dynamic motions (Khansari-Zadeh & Billard, 2011; Ravichandar et al., 2017; Figueroa & Billard, 2018), collaborative behaviors Ewerton et al. (2015); Rozo et al. (2016), and contact-rich manipulation Lin et al. (2012); Abu-Dakka et al. (2018), among others. Often, skills learned from demonstrations need to be refined – due to imperfect data – or adapted to comply with new task requirements. In this context, existing adaptation strategies for GMM policies either build a kernel method on top of the original skill model Huang et al. (2019), or leverage reinforcement learning (RL) to adapt the policy itself (Arenz et al., 2020; Nematollahi et al., 2022). However, none of these techniques explicitly considered the structure of the GMM policy.

Unlike the aforementioned approaches, we propose a policy optimization technique that explicitly considers the underlying GMM structure. To do so, we exploit optimal transport theory (Santambrogio, 2015; Peyré & Cuturi, 2019), which allows us to view the set of GMM policies as a particular space of probability distributions $\mathrm{GMM}_d$. Specifically, we leverage the idea of Chen et al. (2019) and Delon & Desolneux (2020) to view a GMM as a set of discrete measures (dirac masses) on the space of Gaussian distributions $\mathcal{G}(\mathbb{R}^d)$, which is endowed with a Wasserstein distance (see § 2). This allows us to formulate the policy optimization as a Wasserstein gradient flow (WGF) over the space of GMMs (as illustrated in Fig. 1 and explained in §3), where the policy updates are naturally guaranteed to be GMMs. Moreover, we take advantage of the geometry of the Bures-Wasserstein manifold to optimize the Gaussian distributions of a GMM policy via Riemannian optimization. We evaluate our approach over a set of different GMM policies featuring common robot skills: Reaching motions, collision-avoidance behaviors and multi-goal tasks (see § 4). Our results show that our method outperforms common policy optimization baselines in terms of task success rate while providing low-variance solutions.

**Related Work:** Richemond & Maginnis (2017) pioneered the idea of understanding policy optimization through the lens of optimal transport. They interpreted the policy iteration as gradient flows by leveraging the implicit Euler scheme under a Wasserstein distance (see § 2), considering only 1-step return settings. They observed that the resulting policy optimization resembles the gradient flow of the Fokker-Planck equation (JKO scheme) (Jordan et al., 1996). In a similar spirit, Zhang et al. (2018) proposed to use WGFs to formulate policy optimization as a sequence of policy updates traveling along a gradient flow on the space of probability distributions until convergence. To solve the WGF problem, the authors proposed a particle-based algorithm to approximate continuous density functions and subsequently derived the gradients for particle updates based on the JKO scheme. Although Zhang et al. (2018) considered general parametric policies, their method assumed a distribution over the policy parameters and did not consider a specific policy structure, which partially motivated their particle-based approximation. Recently, Mokrov et al. (2021) tackled the computational burden of particle methods by leveraging input-convex neural networks to approximate the WGFs computation. They reformulated the well-known JKO optimization Jordan et al. (1996) over probability measures by an optimization over convex functions. Yet, this work remains a general solution for WFG computation and it did not address its use for policy optimization problems.

Aside from optimal transport approaches, Arenz et al. (2020) proposed a trust-region variational inference for GMMs to approximate multimodal distributions. Although not originally designed for policy optimization, the authors elucidated a connection to learn GMMs of policy parameters in black-box RL. However, their method cannot directly be applied to our GMM policy adaptation setting, nor does it consider the GMM structure from an optimal transport perspective. Nematollahi et al. (2022) proposed SAC-GMM, a hybrid model that employs the well-known SAC algorithm (Haarnoja et al., 2018) to refine dynamic skills encoded by GMMs. The SAC policy was designed to learn residuals on a single vectorized stack of GMM parameters, thus fully disregarding the GMM structure and the geometric constraints of its parameters. Finally, two recent

works share our idea of leveraging geometry in policy optimization: First, a Riemannian proximal policy optimization for GMMs was proposed by Wang et al. (2020), where the geometry induced by the GMM parameters was considered in the optimization via Riemannian gradients, similarly to our method. The policy optimization was regularized by a Wasserstein distance to control the exploration-exploitation trade-off. However, their method did not formulate the policy optimization as an optimal transport problem, i.e. the policy updates do not follow a WGF, as in our approach, but it employed instead a classical non-convex optimization. Second, Moskovitz et al. (2021) employed the Wasserstein natural gradient to exploit the local geometry induced by the Wasserstein regularization of behavioral policy optimization (Pacchiano et al., 2020). In contrast, our method exploits the geometry induced by the structure of the space of GMM policies via the Bures-Wasserstein manifold, which naturally guarantees that policy updates stay on $\mathrm{GMM}_d$.

## 2 BACKGROUND

### 2.1 WASSERSTEIN GRADIENT FLOWS

In Euclidean space a gradient flow is a smooth curve $x : \mathbb{R} \to \mathbb{R}^d$ that satisfies the partial differential equation (PDE) $\dot{x}(t) = -\nabla L(x(t))$ for a given loss function $L : \mathbb{R}^d \to \mathbb{R}$ and starting point $x_0$ at $t = 0$ (Santambrogio, 2015; 2017). A solution can be found straightforwardly by forward discretization, leading to the well-known explicit Euler update scheme $x_{k+1}^\tau = x_k - \lambda \nabla L(x_k^\tau)$, where $\lambda$ denotes the learning rate and $x^\tau$ indicates a discretization of the curve $x(t)$ with discretization parameter $\tau$. Alternatively, we can use a backward discretization, which leads to the following implicit Euler scheme

$$x_{k+1}^\tau = \arg\min_x \left( \frac{\|x - x_k^\tau\|^2}{2\tau} + L(x) \right). \tag{1}$$

Eq. 1 is sometimes referred to as Minimizing Movement Scheme and can be used as an alternative characterization of a gradient flow.

This characterization is particularly interesting when we need to extend the concept of gradient flows to (non-Euclidean) general metric settings, since there is no notion of $\nabla L$ in these cases (Santambrogio, 2015; Ambrosio et al., 2005). Eq. 1 does not involve any gradients and can be expressed using only metric quantities. In this work, we are particularly interested in gradient flows in the $L^2$-Wasserstein space, defined as the set of probability measures $\mathbb{P}(X)$ on a separable Banach space $X$ (Panaretos & Zemel, 2020) and endowed with the $L^2$-Wasserstein distance $W_2$ defined as

$$W_2(\mu, \nu) = \left( \inf_{\gamma \in \Pi(\mu, \nu)} \int_{X \times X} \|x_1 - x_2\|^2 \mathrm{d}\gamma(x_1, x_2) \right)^{\frac{1}{2}}, \tag{2}$$

where $\mu, \nu \in \mathbb{P}(X)$ and $\gamma \in \mathbb{P}(X^2)$ is defined to have the two marginals $\mu$ and $\nu$.

A Generalized Minimizing Movement scheme characterizing gradient flows in the Wasserstein space can be written in analogy to Eq. 1 as:

$$\pi_{k+1}^\tau = \arg\min_\pi \left( \frac{W_2^2(\pi, \pi_k^\tau)}{2\tau} + \mathcal{L}(\pi) \right), \tag{3}$$

where $\mathcal{L}$ is a functional to be minimized on the Wasserstein space and $\pi_k \in \mathbb{P}(X)$. In the following, we will omit the superscript $\tau$ for notational convenience.

### 2.2 REINFORCEMENT LEARNING AS WASSERSTEIN GRADIENT FLOWS

Our view of the policy structure-aware optimization builds on the approach outlined by Richemond & Maginnis (2017), which in turn is based on the JKO scheme of Jordan et al. (1996). They proposed a formulation of 1-step RL problems in terms of Wasserstein gradient flows. In particular, they studied the evolution of a policy $\pi$ under the influence of a free energy functional $J$ of the form:

$$J(\pi) = K_r(\pi) + \beta \mathcal{H}(\pi) = \int_A \mathrm{d}\pi(\boldsymbol{a}|\boldsymbol{s}) r(\boldsymbol{s}, \boldsymbol{a}) - \beta \int_A \mathrm{d}\pi(\boldsymbol{a}|\boldsymbol{s}) \log(\pi(\boldsymbol{a}|\boldsymbol{s})), \tag{4}$$

where $K_r(\pi)$ denotes the inner energy of the system, here determined by the reward $r(\boldsymbol{s}, \boldsymbol{a})$. Moreover, $\mathcal{H}(\pi)$ is the entropy of the policy $\pi(\boldsymbol{a}|\boldsymbol{s})$, with $\boldsymbol{s}$ and $\boldsymbol{a}$ denoting the state and action, respectively. Thus Eq. 4 can be recognized as the usual objective in 1-step RL settings with entropy regularization. It is well known that the evolution of probability densities under a free energy of this form is properly described by a PDE known as the Fokker-Planck equation. Richemond & Maginnis (2017) exploited the result of Jordan et al. (1996), which stated that this evolution can be interpreted as the gradient flow of the functional $J$ in Wasserstein space. This flow is characterized by the following minimizing movement scheme

$$\pi_{k+1} = \arg\min_\pi \left( \frac{W_2^2(\pi, \pi_k)}{2\tau} - J(\pi) \right), \tag{5}$$

which naturally provides iterative updates for the policy $\pi$. While Richemond & Maginnis (2017) considered a 1-step bandit setting, we extend this approach to full multi-step RL problems and learn policies for long-horizon tasks.

## 2.3 THE $L^2$-WASSERSTEIN DISTANCE BETWEEN GAUSSIAN MIXTURE MODELS (GMMs)

In this paper, we consider policies $\pi(\boldsymbol{x})$ that build on a GMM structure, i.e., $\pi(\boldsymbol{x}) = \sum_{i=1}^N \omega_i \mathcal{N}(\boldsymbol{x}; \boldsymbol{\mu}_i, \boldsymbol{\Sigma}_i)$, where $\mathcal{N}$ denotes a multivariate Gaussian distribution with mean $\boldsymbol{\mu}_i$ and covariance matrix $\boldsymbol{\Sigma}_i$, and $\omega_i$ are the weights of the $N$ individual Gaussian components, which are subject to $\sum_i \omega_i = 1$. In the following, we will write $\hat{\boldsymbol{\mu}}$, $\hat{\boldsymbol{\Sigma}}$ and $\hat{\boldsymbol{\omega}}$ to denote the stacked means, covariance matrices and weights of the $N$ components. Therefore, we do not consider WGFs on the full manifold of probability distributions (Wasserstein space) $\mathbb{P}(\mathbb{R}^d)$ but rather focus on WGFs evolving on the submanifold of GMMs, that is $\mathrm{GMM}_d \subset \mathbb{P}(\mathbb{R}^d)$. Following Chen et al. (2019); Delon & Desolneux (2020), we can approximately describe this submanifold as a discrete distribution over the space of Gaussian distributions equipped with the Wasserstein metric. This in turn can be identified with the Bures-Wasserstein manifold which is the product manifold $\mathbb{R}^d \times \mathcal{S}_{++}^d$, where $\mathcal{S}_{++}^d$ denotes the Riemannian manifold of $d$-dimensional symmetric positive definite matrices. The corresponding approximated Wasserstein distance between two GMMs $\pi_1$, $\pi_2$ is given by

$$W_2^2\big(\pi_1(\boldsymbol{x}), \pi_2(\boldsymbol{x})\big) = \min_{\boldsymbol{P} \in U(\omega_1, \omega_2)} \sum_{i,j}^N \boldsymbol{P}_{ij} W_2^2\big(\mathcal{N}_1(\boldsymbol{x}; \boldsymbol{\mu}_i, \boldsymbol{\Sigma}_i), \mathcal{N}_2(\boldsymbol{x}; \boldsymbol{\mu}_j, \boldsymbol{\Sigma}_j)\big), \tag{6}$$

where $U(\omega_1, \omega_2) = \{\boldsymbol{P} \in \mathbb{R}_+^{N \times N} | \boldsymbol{P}\mathbf{1}_N = \omega_1, \boldsymbol{P}^\mathsf{T}\mathbf{1}_N = \omega_2\}$ with $\mathbf{1}_N$ denoting an $N$-dimensional vector of ones. The Wasserstein distance between two Gaussian distributions in Eq. 6 can be computed analytically as follows

$$W_2^2\big(\mathcal{N}_1(\boldsymbol{x}; \boldsymbol{\mu}_i, \boldsymbol{\Sigma}_i), \mathcal{N}_2(\boldsymbol{x}; \boldsymbol{\mu}_j, \boldsymbol{\Sigma}_j)\big) = \|\boldsymbol{\mu}_i - \boldsymbol{\mu}_j\|^2 + \mathrm{tr}\left[ \boldsymbol{\Sigma}_i + \boldsymbol{\Sigma}_j - 2\left( \boldsymbol{\Sigma}_i^{1/2} \boldsymbol{\Sigma}_j \boldsymbol{\Sigma}_i^{1/2} \right) \right]. \tag{7}$$

## 2.4 LEARNING GMM POLICIES FROM DEMONSTRATIONS

A popular approach in RL – particularly in the robotics domain – to reduce the number of policy rollouts in the environment is to warm-start the policy with a set of demonstrations provided by an expert. In this work we choose to represent our policy via a GMM. We assume that demonstrations are provided as a set of trajectories $\tau$ of state-action pairs $\tau = \{(\boldsymbol{s}_0, \boldsymbol{a}_0), (\boldsymbol{s}_1, \boldsymbol{a}_1), \dots (\boldsymbol{s}_T, \boldsymbol{a}_T)\}$. To initialize our policy, we first use the Expectation-Maximization (EM) algorithm to fit a GMM, in the joint state-action space, to the demonstrations. This results in a mixture distribution $\pi(\boldsymbol{s}, \boldsymbol{a}) = \sum_{i=1}^N \omega_i \mathcal{N}\big([\boldsymbol{s}\,\boldsymbol{a}]^\mathsf{T}; \boldsymbol{\mu}_i, \boldsymbol{\Sigma}_i\big)$ from which a policy can be obtained by conditioning on the state $\boldsymbol{s}$, as follows

$$\pi(\boldsymbol{a}|\boldsymbol{s}) = \frac{\pi(\boldsymbol{s}, \boldsymbol{a})}{\int \pi(\boldsymbol{s}, \boldsymbol{a})\mathrm{d}\boldsymbol{a}}. \tag{8}$$

In the context of GMMs, this is also known as Gaussian Mixture Regression (GMR) (Ghahramani & Jordan, 1994). The resulting conditional distribution is another GMM on action space with state-dependent parameters, given by

$$\pi(\boldsymbol{a}_t|\boldsymbol{s}_t) = \sum_{i=1}^N \omega_i(\boldsymbol{s}_t)\mathcal{N}(\boldsymbol{a}_t; \boldsymbol{\mu}_i^a(\boldsymbol{s}_t), \boldsymbol{\Sigma}_i^a). \tag{9}$$

Details on computation of Eq. 9 from the original GMM are given in App. A.1.

## 3 WASSERSTEIN GRADIENT FLOWS FOR GMM POLICY OPTIMIZATION

In this work, we focus on multi-step RL tasks for policy adaptation. We consider a finite-horizon Markov Decision Process (MDP) with continuous state and action spaces $\mathcal{S} \in \mathbb{R}^n$ and $\mathcal{A} \in \mathbb{R}^m$, transition and reward functions $p(\boldsymbol{s}_{t+1}|\boldsymbol{s}_t, \boldsymbol{a}_t)$ and $r(\boldsymbol{s}_t, \boldsymbol{a}_t)$, initial state distribution $\rho(\boldsymbol{s}_0)$ and a discount factor $\gamma$. Further, we assume to have an initial policy $\pi(\boldsymbol{a}_t|\boldsymbol{s}_t)$, which is to be adapted by optimizing some objective function $K_r(\pi)$. As stated in § 1, this problem arises in robot learning settings where a policy learned via imitation learning (e.g., LfD) needs to be adapted to new objectives or unseen environmental conditions. To promote exploration and avoid premature convergence to suboptimal policies, we leverage maximum entropy RL (Eysenbach & Levine, 2022) by adding an entropy term $\mathcal{H}(\pi)$ to the objective. Thus, the overall objective has the form of a free energy functional (resembling Eq. 4) and can be written as

$$J(\pi) = K_r(\pi) + \beta \mathcal{H}(\pi), \tag{10}$$

where $\beta$ is a hyperparameter and $K_r(\pi)$ corresponds to the usual cumulative return

$$K_r(\pi) = \mathbb{E}_\tau \left[ \sum_t r(\boldsymbol{s}_t, \boldsymbol{a}_t) \right] = \int \Pi_t \mathrm{d}\boldsymbol{s}_0 \mathrm{d}\boldsymbol{s}_t \mathrm{d}\boldsymbol{a}_t \rho(\boldsymbol{s}_0) \pi(\boldsymbol{a}_t|\boldsymbol{s}_t) p(\boldsymbol{s}_{t+1}|\boldsymbol{s}_t, \boldsymbol{a}_t) \sum_t \gamma^t r(\boldsymbol{s}_t, \boldsymbol{a}_t). \tag{11}$$

The evolution of the policy $\pi(\boldsymbol{a}_t|\boldsymbol{s}_t)$ over the course of the optimization can be described as a flow of a probability distribution in Wasserstein space. This formulation comes with three major benefits: *(i)* We directly leverage the Wasserstein metric properties for describing the evolution of probability distributions; *(ii)* We exploit the $L^2$-Wasserstein distance to constrain the policy updates, which is important to guarantee stability in policy optimization (Schulman et al., 2015; 2017; Otto et al., 2021); *(iii)* By constraining to specific submanifolds of the Wasserstein space, in this case GMMs, we can impose additional structural properties on the policy optimization.

Since our objective in Eq. 10 has the form of the free energy functional studied by Richemond & Maginnis (2017); Jordan et al. (1996), we can leverage the iterative updates scheme of Eq. 5 to formulate the evolution of our policy iteration under the flow generated by Eq. 10. As mentioned previously, we focus on the special case of GMM policies and therefore restrict the Wasserstein gradient flow to the submanifold of GMM distributions $\mathrm{GMM}_d$. We refer the interested reader to App. A.3, where we provide the explicit form of $J(\pi)$ of Eq. 10 for the GMM case.

### 3.1 POLICY OPTIMIZATION

To begin with, we leverage the approximation that describes the GMM submanifold as a discrete distribution over the space of Gaussian distributions $\mathcal{G}(\mathbb{R}^d)$, equipped with the Wasserstein metric (Chen et al., 2019). Consequently, our policy optimization problem naturally splits into an optimization over the $(N-1)$-dimensional simplex and an optimization on the $N$-fold product of the Bures-Wasserstein manifold ($BW^N$), i.e. the product manifold $\left( \mathbb{R}^d \times \mathcal{S}_{++}^d \right)^N$. The former corresponds to the GMM weights while the latter applies to the set of Gaussian distributions parameters. Note that the identification with the $BW^N$ manifold allows us to perform the optimization directly on the parameter space. This comes with several benefits: *(i)* We can leverage the well-known analytic solution of the Wasserstein distance between two Gaussian distributions in Eq. 6, greatly reducing the computational complexity of the policy optimization. *(ii)* As Chen et al. (2019) show, we can guarantee that the policy optimized via Eq. 6 remains a GMM. *(iii)* Unlike the full Wasserstein space[1], the resulting product manifold is a true Riemannian manifold such that we can leverage the machinery of Riemannian optimization. Importantly, working in the parameter space allows us to apply an explicit Euler scheme instead of the implicit formulation of Eq. 3.

According to the above-mentioned split, we formulate the policy optimization as an EM-like two-step procedure that alternates between the Gaussian parameters (i.e. means and covariance matrices) and the GMM weights. To optimize the former, we propose to leverage the Riemannian structure of the $BW$ manifold to reformulate the updates as a forward discretization, similarly to Chen & Li (2020). In other words, by embedding the Gaussian components of the GMM policy in a Riemannian manifold, the Wasserstein gradient flow in the implicit form of Eq. 5 can be approximated

---

[1] Wasserstein space is not a true Riemannian manifold, but it can be equipped with a Riemannian structure and formal calculus on this manifold (Otto, 2001), which has been made rigorous by (Ambrosio et al., 2005)

by an explicit Euler update scheme according to the $BW$ metric (further details are provided in App. A.4). This allows us to leverage the expressions of the Riemannian gradient and exponential map of the $BW$ manifold (Malagò et al., 2018; Han et al., 2021). Thus, the optimization boils down to Riemannian gradient descent where the gradient is defined w.r.t the Bures-Wasserstein metric. In particular, we use the expression for Riemannian gradient, metric and exponential map used in (Han et al., 2021). Formally, the resulting updates for the Gaussian parameters of the GMM follow the Riemannian gradient descent scheme given by:

$$\hat{\boldsymbol{\mu}}_{k+1} = \mathrm{R}_{\hat{\boldsymbol{\mu}}_k}\left(\lambda \cdot \mathrm{grad}_{\hat{\boldsymbol{\mu}}} \, J(\pi_k)\right), \quad \text{and} \quad \hat{\boldsymbol{\Sigma}}_{k+1} = \mathrm{R}_{\hat{\boldsymbol{\Sigma}}_k}\left(\lambda \cdot \mathrm{grad}_{\hat{\boldsymbol{\Sigma}}} \, J(\pi_k)\right), \quad (12)$$

where grad denotes the Riemannian gradient w.r.t. the Bures-Wasserstein metric, $\mathrm{R}_{\boldsymbol{x}} : \mathcal{T}_{\boldsymbol{x}}\mathcal{M} \to \mathcal{M}$ denotes the retraction operator, which maps a point on the tangent space $\mathcal{T}_{\boldsymbol{x}}\mathcal{M}$ back to the manifold $\mathcal{M} \equiv \mathrm{BW}$ (Boumal, 2022). Moreover, $\lambda$ is a learning rate and $\pi_k \stackrel{\text{def}}{=} \pi(\hat{\boldsymbol{\mu}}_k, \hat{\boldsymbol{\Sigma}}_k, \hat{\boldsymbol{\omega}}_k)$. The Euclidean gradients of $J(\pi)$ required for computing grad can be obtained using a likelihood ratio estimator (a.k.a score function estimator or REINFORCE) (Williams, 2004) and are provided in App. A.3.

Concerning the GMM weights, we first reparameterize them as $\omega_j = \frac{\exp \eta_j}{\sum_{k=1}^{N} \exp \eta_k}$ and optimize w.r.t. the new parameters $\eta_j, j = 1...N$, which unlike $\hat{\boldsymbol{\omega}}$ are unconstrained. For this optimization we employ the implicit Euler scheme:

$$\hat{\boldsymbol{\eta}}_{k+1} = \arg\min_{\hat{\boldsymbol{\eta}}} \left(\frac{W_2^2(\pi_{k+1}(\hat{\boldsymbol{\eta}}), \pi_k)}{2\tau} - J(\pi_{k+1}(\hat{\boldsymbol{\eta}}))\right), \quad (13)$$

where $\pi_{k+1}(\hat{\boldsymbol{\eta}}) \stackrel{\text{def}}{=} \pi(\hat{\boldsymbol{\mu}}_{k+1}, \hat{\boldsymbol{\Sigma}}_{k+1}, \hat{\boldsymbol{\eta}})$. We minimize Eq. 13 by gradient descent w.r.t. $\eta$ as follows:

$$\hat{\boldsymbol{\eta}}_{k+1} = \hat{\boldsymbol{\eta}}_k - \lambda \nabla_{\hat{\boldsymbol{\eta}}} \left(\frac{W_2^2(\pi_{k+1}(\boldsymbol{\eta}), \pi_k)}{\tau} - J(\pi_{k+1}(\hat{\boldsymbol{\eta}}))\right). \quad (14)$$

The gradient of $J(\pi)$ can be obtained analytically using a likelihood ratio estimator. For the Wasserstein term, we first compute the gradient w.r.t. the weights via the Sinkhorn algorithm (Cuturi & Doucet, 2014), from which the gradient w.r.t $\boldsymbol{\eta}$ can be then obtained via the chain rule. Note that we have to rely on the Sinkhorn algorithm here since there is no analytic solution available for the Wasserstein distance between discrete distributions, unlike the above case of the Gaussian components. Consequently, we cannot compute the corresponding gradients.

### 3.2 IMPLEMENTATION PIPELINE

To carry out the policy optimization, we proceed as in the usual on-policy RL scheme: We first roll out the current policy to collect samples of state-action-reward tuples. Then, we use the collected interaction trajectories to compute a sample-based estimate of the functional $K_r(\pi)$ and its gradients w.r.t the policy parameters, as explained in § 3.1. An optimization step consists of alternating between optimizing the Gaussian parameters using 12, and updating the weights via 14. For the optimization of the Gaussian parameters we leverage Pymanopt (Townsend et al., 2016) for Riemannian optimization. We extended this library by implementing the Bures-Wasserstein manifold based on the expressions provided by Han et al. (2021) (see App. A.2 for details). Furthermore, we added a custom line-search routine that accounts for a constraint on the Wasserstein distance between the old and the optimized GMM, as to our knowledge such a search method does not exist in out-of-the-box optimizers. The details of this custom line-search can be found in Algorithm 2 in App. A.5. Regarding the optimization of the GMM weights, we use POT (Flamary et al., 2021), a Python library for optimal transport, from which we obtain the quantities required for computing the gradients of the Wasserstein distance w.r.t. the weights in Eq. 14.

The full policy optimization finishes if either the objective stops improving or the Wasserstein distance between the old and optimized GMMs exceeds a predefined threshold, which is chosen experimentally. Afterwards, fresh rollouts are performed with the updated policy and the aforementioned two-step procedure starts over. This optimization loop is repeated until a task-dependent success criterion has been fulfilled. We summarize the proposed optimization in Algorithm 1.

## 4 EXPERIMENTS

We tested our approach in three different robotic settings: a reaching skill, a collision-free trajectory tracking, and a multiple-goal task. All the tasks are represented in a 2D operational space. The robot

---

**Algorithm 1** GMM Policy Optimization via Wasserstein Gradient Flows

---
    **Input**: initial policy $\pi(\boldsymbol{a}|\boldsymbol{s})$
1: **while** not goal reached **do**
2:    Rollout policy $\pi(\boldsymbol{a}|\boldsymbol{s})$ in the environment for $M$ episodes to collect interaction trajectories $\tau = \{(\boldsymbol{s}_0, \boldsymbol{a}_0, \boldsymbol{r}_0), (\boldsymbol{s}_1, \boldsymbol{a}_1, \boldsymbol{r}_1), \ldots, (\boldsymbol{s}_T, \boldsymbol{a}_T, \boldsymbol{r}_T)\}_{m=1}^{M}$
3:    **repeat**
4:        Update Gaussian components parameters $\hat{\boldsymbol{\mu}}$, $\hat{\boldsymbol{\Sigma}}$ using Riemannian optimization (12), where $\lambda^{ls}$ is determined via line-search (see §3.2).
5:    **until** convergence
6:    **repeat**
7:        Update GMM weights $\hat{\boldsymbol{\omega}}$ via gradient descent on the free energy objective 10, using 14
8:    **until** converged
9: **end while**

---

motion policies were initially learned from human demonstrations collected on a simple Python graphical interface. We assumed we were given $M$ demonstrations, each of which contained $T_m$ data points for a dataset of $N = \sum_m T_m$ total observations $\tau = \{(\boldsymbol{s}_t, \boldsymbol{a}_t)\}_{t=1}^{N}$. The state $\boldsymbol{s}$ and action $\boldsymbol{a}$ correspond to the robot end-effector position $\boldsymbol{x} \in \mathbb{R}^2$ and velocity $\dot{\boldsymbol{x}} \in \mathbb{R}^2$. The GMM models were trained via classical Expectation-Maximization. The policy rollout consists of sampling a velocity action $\boldsymbol{a}_t \sim \pi(\boldsymbol{a}_t|\boldsymbol{s}_t)$ using Eq. 9, and subsequently commanding the robot via a Cartesian velocity controller at a frequency of 100Hz. For all the experiments, we used the Robotics Toolbox for Python (Corke & Haviland, 2021) to simulate the robotic environments.

To show the importance of accounting for the policy structure in RL settings, we compared our method against two structure-unaware baselines: PPO (Schulman et al., 2017) and SAC-GMM (Nematollahi et al., 2022). As PPO was not originally designed to directly optimize the parameters of a previously-learned GMM policy, we designed the policy actions to represent (usually small) corrections to the GMM parameters, i.e. $\boldsymbol{a} = [\Delta\boldsymbol{\omega} \; \Delta\hat{\boldsymbol{\mu}} \; \Delta\,\text{vec}(\hat{\boldsymbol{\Sigma}})]$, following the same methodology as SAC-GMM (Nematollahi et al., 2022). The PPO and SAC implementations correspond to the code provided by Stable-Baselines3 (Raffin et al., 2021), whose policies are parametrized by MLP networks. During policy optimization, we sample an action from the MLP policy that is then used to update the GMM parameters by adding the computed corrections to the current parameters. Later, we proceed as described earlier, namely, the updated GMM is used to compute the velocity action via Eq. 9. For comparison purposes, we report statistical results for the three considered settings over 5 runs for the task success rate and solution variance. We tuned the baselines separately for each task using Optuna (Akiba et al., 2019). In addition, to assess the importance of our Riemannian formulation, we performed an ablation where we used the implicit scheme based on Euclidean gradient descent instead of the explicit optimization on the Bures-Wasserstein manifold (see App. A.6.2). Last but not least, we tested our approach on a 3D version of the collision-free task performed by a 7-DoF Franka Emika Panda robot in a virtual environment as reported in App. A.6.3.

## 4.1 TASKS DESCRIPTION

**Reaching Task:** This experiment consists of: (1) learning an initial GMM policy such that the robot end-effector reaches a target by following an L-shape trajectory from its initial position, and (2) adapting the GMM policy to reach a new target located midway and above the previously-learned L-shape trajectories. The initial policy, shown in Fig. 2-*left* and Fig. 12a, was learned from 12 demonstrations and encoded by a 7-component GMM. To adapt the policy, we defined a dense reward as a function of the position error between the robot end-effector and the new target. We also added a sparse penalty term that punishes rollouts leading to significantly divergent trajectories. Convergence is achieved when a minimum average position error w.r.t the target – computed over an episode – is reached.

**Collision-avoidance Task:** This task consists of: (1) learning an initial GMM policy of a linear reaching motion, and (2) adapting the GMM policy to reach a new horizontally-translated target while avoiding to collide with two spherical obstacles located midway between the initial robot position and the new target. The initial GMM policy was learned from 10 human demonstrations and represented by a 3-component GMM, as shown in Fig. 2-*middle* and Fig. 12b. For policy optimization, we defined a sparse reward as a function of the position error between the robot end-effector

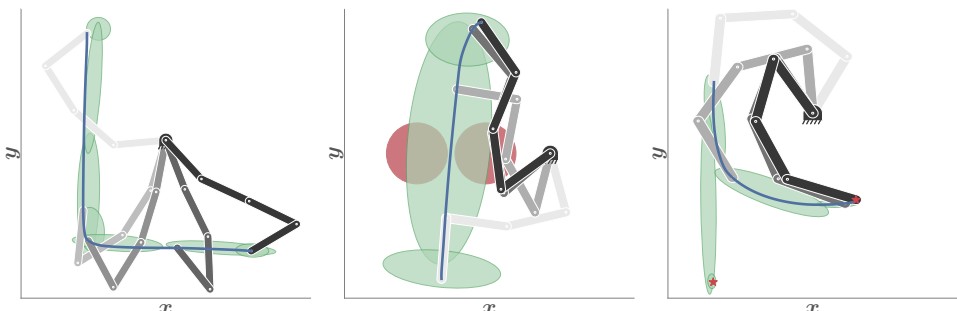

Figure 2: The three tested robotic settings: a reaching skill (*left*), a collision-free trajectory tracking (*middle*), and a multiple-goal task (*right*). The robot color goes from light gray to black to show the evolution of the task reproduction. Green Gaussian components (⬭) depict the initial GMM policy, projected on the 2D Cartesian position space. The end-effector trajectory resulting from the initial GMM policy is shown in dark blue lines (—). Red circles (⬤) in the collision-avoidance task represent the obstacles (*middle*). The different targets of the multiple-goal task (*right*) are depicted as red stars.

position and the target at the end of the rollout. We also included two sparse penalty terms: the first one punishes rollouts leading to collisions with the obstacles, for which the rollout is stopped; the second term penalizes rollouts with significantly divergent trajectories. Convergence is determined by a minimum average position error w.r.t the target computed over an episode.

**Multiple-goal Task:** This setting involves: (1) learning an initial GMM policy where the robot end-effector reaches two different targets (i.e., task goals) starting from the same initial position, and (2) adapting the initial policy to reach a new target located close to one of the previous task goals. The intended adapted behavior should make the robot go through the most relevant GMM components according to the new target location. The initial GMM policy was learned from 12 demonstrations and encoded by a 6-component GMM, as shown in Fig. 2-*right* and Fig. 12c. To optimize the initial GMM policy, we specified a sparse reward based on the position error between the robot end-effector position and the chosen target at the end of the rollout. Similar to the previous experiments, we added a sparse penalty term to penalize rollouts generating significantly divergent trajectories. Again, the policy optimization converges when the average position error w.r.t the chosen target reaches a minimum threshold.

## 4.2 Results Analysis

The reaching task tested our method's ability to adapt a previously-learned reaching skill to a new goal, located at $(6.0, -6.5)$ (cf. Fig. 12a-*left*). Achieving this required to adapt the Gaussian parameters of mainly the last four GMM components, while the other ones remained unchanged. We compared all methods in terms of the success rate over environment steps, where the success rate is defined as the percentage of rollouts that reach the new goal. Figure 3-*left* shows that our method achieved a success rate of 1 after approximately 70000 environment interactions. Despite PPO was also able to complete the task reliably, it required many more environment steps (cf. Fig. 5-*left*). In sharp contrast, SAC did not reach any improvement. These observations underline the importance of some kind of trust region or constraint on the policy updates, which allowed both our method and PPO to reach good success rates. Furthermore, this experiment showed that our method is much more sample-efficient in adapting the GMM parameters, which we attribute to the fact that our method explicitly takes the GMM structure into account in the formulation of the optimization.

In the collision-avoidance task, we tested whether our method was able to adapt a trajectory tracking skill in order to avoid collisions with newly added obstacles. These were placed in such a way that the robot was forced to move its end-effector through a narrow path between the obstacles (cf. Fig. 2-*middle*). While the reaching task could be adapted by mainly varying the means of the GMM components, this task also demands to adapt the covariance of the second GMM component. Figure 3-*middle* shows that our method solved this task reliably after comparatively few environment interactions. Although PPO also achieved a success rate of 1, it took 6 times more environment steps than our method. SAC only reached an average success rate of $0.8$, however with high variance (cf. Fig. 5-*middle*). These results again show the importance of the constraints on the policy updates. The huge discrepancy in the required environment steps between our method and PPO further emphasizes the importance of taking the GMM structure into account in the policy optimization.

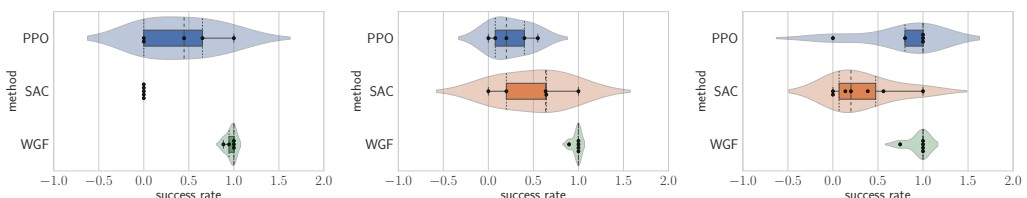

Figure 3: Success rate of our method (WGF) and the baselines on the reaching (*left*), the collision-avoidance (*middle*) and the multiple-goal tasks (*right*). The shaded area depicts the standard deviation over 5 runs.

Figure 4: Variance of the success rate over the 5 runs for our method (WGF) and the two baselines on the reaching task (*left*), the collision avoidance task (*middle*) and the multiple-goal task (*right*). The violine plots are overlaid with box plots, quartile lines and a swarm plot, where dots indicate the success rates of individual runs. The plots show the variance at the following time steps from left to right: 80000, 90000, 95000.

While the previous two tasks were accomplished by adapting mostly the Gaussian parameters of the GMM, the multiple-goal task requires to adapt the GMM weights. The initial skill comprised reaching motions to two different goals and an execution of this skill results in reaching one of them, depending on the sampling noise (cf. Fig. 12c). The easiest way to adapt the policy to reach only one of the two goals is to reduce the GMM weights of the components belonging to the undesired motion and correspondingly increase the weights of the other components. As shown in Fig. 3-*right*, our method again quickly achieved a success rate of 1. PPO required substantially many more environment steps, while SAC was not able to solve the task.

In Fig. 4 we report the success rate variance over 5 runs at a fixed time step, which corresponded to the step at which the first method achieved a success rate of 1, thus prioritizing sample efficiency. The plots show that our method exhibits a very low solution variance. Both baselines varied largely, except for the reaching task, where all SAC runs collapse to a success rate of 0. These results show that our method, despite showing large variance at the start, was able to quickly reduce the variance and converge reliably to a good success rate. We also provide similar plots of solution variance in Fig. 6, where we report the results for each method using its own convergence time step.

## 5   CONCLUSIONS AND FUTURE WORK

We presented a novel method for GMM policy optimization, which leverages optimal transport theory to formulate the policy optimization as a Wasserstein gradient flow on the manifold of GMMs. Our formulation explicitly accounts for the GMM structure in the optimization and furthermore enables us to naturally constrain the policy updates by the $L^2$-Wasserstein distance between GMMs to enhance the stability of the policy optimization process. Moreover, the embedding of the Gaussian components of the GMM policy in the Bures-Wassertein manifold greatly reduced the computational cost of the policy optimization. Experiments on three robotic tasks provided strong evidence of the importance of our policy-structure aware optimization against approaches that disregard the GMM structure. A possible limitation of our method is that each optimization loop involves running the Sinkhorn algorithm, which is computationally expensive. This might be improved by employing recent advances on initializing the Sinkhorn algorithm (Thornton & Cuturi, 2022). Also, we observed an intricate interplay between the optimization of the GMM weights and the Gaussian parameters, which sometimes resulted in one update hampering the other. In future work we plan to address the latter problem by using separate adaptive learning rates for weights and Gaussian parameters. Another possibility would entail to reformulate the approach as a fully dynamical, particle-based optimization on the Bures-Wasserstein manifold, where both the locations and weights of the particles are updated using Wasserstein Fisher-Rao gradient flows Chizat et al. (2015); Chizat (2019); Liero et al. (2018). Finally it would be interesting to combine our method with an actor-critic formulation and to replace the multi-step cumulative reward by a trained $Q$-function.

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

# A APPENDIX

## A.1 DETAILS ON GAUSSIAN MIXTURE REGRESSION (GMR)

In GMR we start from a GMM in state-action space $\pi(\boldsymbol{s}, \boldsymbol{a}) = \sum_{i=1}^{N} \omega_i \mathcal{N}\big([\boldsymbol{s}\,\boldsymbol{a}]^{\mathsf{T}}; \boldsymbol{\mu}_i, \boldsymbol{\Sigma}_i\big)$ from which a policy, i.e. a probability distribution on the action space, can be obtained by conditioning on the state, as follows

$$\pi(\boldsymbol{a}|\boldsymbol{s}) = \frac{\pi(\boldsymbol{s}, \boldsymbol{a})}{\int \pi(\boldsymbol{s}, \boldsymbol{a}) \mathrm{d}\boldsymbol{a}}. \tag{15}$$

The resulting conditional distribution is another GMM on the action sapce, with state dependent parameters, given by:

$$\pi(\boldsymbol{a}_t|\boldsymbol{s}_t) = \sum_{i=1}^{N} \omega_i(\boldsymbol{s}_t) \mathcal{N}(\boldsymbol{a}_t; \boldsymbol{\mu}_i^a(\boldsymbol{s}_t), \boldsymbol{\Sigma}_i^a), \quad \text{with} \tag{16}$$

$$\boldsymbol{\mu}_i^a(\boldsymbol{s}_t) = \boldsymbol{\mu}_i^a + \boldsymbol{\Sigma}_i^{as}(\boldsymbol{\Sigma}_i^s)^{-1}(\boldsymbol{s}_t - \boldsymbol{\mu}_i^s), \tag{17}$$

$$\boldsymbol{\Sigma}_i^a = \boldsymbol{\Sigma}_i^a - \boldsymbol{\Sigma}_i^{as}(\boldsymbol{\Sigma}_i^s)^{-1}\boldsymbol{\Sigma}_i^{sa}, \tag{18}$$

$$\omega_i(\boldsymbol{s}_t) = \frac{\omega_i \mathcal{N}(\boldsymbol{s}_t; \boldsymbol{\mu}_i^s, \boldsymbol{\Sigma}_i^s)}{\sum_{k}^{n} \omega_k \mathcal{N}(\boldsymbol{s}_t; \boldsymbol{\mu}_k^s, \boldsymbol{\Sigma}_k^s)}. \tag{19}$$

Note that we have split the GMM parameters $\boldsymbol{\mu}_i$ and $\boldsymbol{\Sigma}_i$ into their state and action components according to

$$\boldsymbol{\mu}_i = \begin{pmatrix} \boldsymbol{\mu}_i^s \\ \boldsymbol{\mu}_i^a \end{pmatrix}, \quad \boldsymbol{\Sigma}_i = \begin{pmatrix} \boldsymbol{\Sigma}_i^s & \boldsymbol{\Sigma}_i^{sa} \\ \boldsymbol{\Sigma}_i^{as} & \boldsymbol{\Sigma}_i^a \end{pmatrix}. \tag{20}$$

## A.2 RIEMANNIAN GRADIENTS AND RETRACTIONS

For completeness we give here the explicit expressions of the Riemannian gradients and the retractions used in § 3.1. As the mean vectors are assumed to lie in the Euclidean space, their Riemannian gradients actually coincide with the Euclidean gradients and no retraction is required, so Eq. 12 reduces to the well-known Euclidean gradient descent

$$\hat{\boldsymbol{\mu}}_{k+1} = \hat{\boldsymbol{\mu}}_k + \nabla_{\hat{\boldsymbol{\mu}}} J(\pi_k), \tag{21}$$

where $\nabla_{\hat{\boldsymbol{\mu}}}$ denotes the Euclidean gradient w.r.t. $\hat{\boldsymbol{\mu}}$. For the covariance matrices we use the gradient and retraction w.r.t. the Bures-Wasserstein manifold, taken from (Malagò et al., 2018; Han et al., 2021). The gradient is given by

$$\mathrm{grad}_{\hat{\boldsymbol{\Sigma}}} J(\pi_k) = 4\{\nabla_{\hat{\boldsymbol{\Sigma}}} J(\pi_k)\hat{\boldsymbol{\Sigma}}\}_S, \tag{22}$$

where again $\nabla_{\hat{\boldsymbol{\Sigma}}}$ denotes the Euclidean gradient w.r.t. $\hat{\boldsymbol{\Sigma}}$ and $\{\boldsymbol{X}\}_S = \frac{(\boldsymbol{X}+\boldsymbol{X}^{\mathsf{T}})}{2}$. Furthermore, the retraction is given by

$$\mathrm{R}_{\boldsymbol{\Sigma}_k}\left(\hat{\boldsymbol{X}}\right) = \hat{\boldsymbol{\Sigma}}_k + \hat{\boldsymbol{X}} + \mathcal{L}_{\hat{\boldsymbol{X}}}\left[\hat{\boldsymbol{\Sigma}}_k\right]\hat{\boldsymbol{X}}\mathcal{L}_{\hat{\boldsymbol{X}}}\left[\hat{\boldsymbol{\Sigma}}_k\right], \tag{23}$$

where $\mathcal{L}_{\hat{\boldsymbol{X}}}\left[\hat{\boldsymbol{\Sigma}}_k\right]$ is the Lyapunov operator, defined as the solution to the matrix linear system $\mathcal{L}_{\hat{\boldsymbol{X}}}\left[\hat{\boldsymbol{\Sigma}}_k\right]\hat{\boldsymbol{X}} + \hat{\boldsymbol{X}}\mathcal{L}_{\hat{\boldsymbol{X}}}\left[\hat{\boldsymbol{\Sigma}}_k\right] = \hat{\boldsymbol{\Sigma}}_k.$

## A.3 EXPRESSIONS OF THE FREE FUNCTIONAL $J(\pi)$ AND ITS EUCLIDEAN GRADIENTS

For completeness sake, we provide here the explicit expression of the Euclidean gradients for the objective $J(\pi)$ w.r.t. the parameters of the GMM, which are used in the construction of the Riemannian gradients. Using the policy gradient theorem, we obtain the gradient of Eq. 11 w.r.t to a

parameter $\boldsymbol{\xi}$ as follows

$$\nabla_{\boldsymbol{\xi}} J(\pi) = \nabla_{\boldsymbol{\xi}} \int \Pi_t \mathrm{d}\boldsymbol{s}_0 \mathrm{d}\boldsymbol{s}_t \mathrm{d}\boldsymbol{a}_t \rho(\boldsymbol{s}_0) \pi(\boldsymbol{a}_t|\boldsymbol{s}_t) p(\boldsymbol{s}_{t+1}|\boldsymbol{s}_t, \boldsymbol{a}_t) \sum_{t'>t} \gamma^t r(\boldsymbol{s}_{t'}, \boldsymbol{a}_{t'}), \qquad (24)$$

$$= \mathbb{E}_\tau \left[ \sum_t \nabla_{\boldsymbol{\xi}} \log(\pi(\boldsymbol{a}_t|\boldsymbol{s}_t)) \sum_{t'>t} r(\boldsymbol{s}_t, \boldsymbol{a}_t) \right],$$

$$= \mathbb{E}_\tau \left[ \sum_t \nabla_{\boldsymbol{\xi}} \log \left( \frac{\pi(\boldsymbol{s}_t, \boldsymbol{a}_t)}{\int \mathrm{d}\boldsymbol{a}_t \pi(\boldsymbol{s}_t, \boldsymbol{a}_t)} \right) \sum_{t'>t} r(\boldsymbol{s}_t, \boldsymbol{a}_t) \right],$$

$$= \sum_t \mathbb{E}_\tau \left[ \left( \frac{\nabla_{\boldsymbol{\xi}} \pi(\boldsymbol{s}_t, \boldsymbol{a}_t)}{\pi(\boldsymbol{s}_t, \boldsymbol{a}_t)} - \frac{\int \mathrm{d}\boldsymbol{a}_t \nabla_{\boldsymbol{\xi}} \pi(\boldsymbol{s}_t, \boldsymbol{a}_t)}{\int \mathrm{d}\boldsymbol{a}_t \pi(\boldsymbol{s}_t, \boldsymbol{a}_t)} \right) \sum_{t'>t} r(\boldsymbol{s}_t, \boldsymbol{a}_t) \right].$$

In this work, we focus on GMM policies, for which the objective $J(\pi)$ takes the form:

$$J(\pi) = \int \Pi_t \mathrm{d}\boldsymbol{s}_0 \mathrm{d}\boldsymbol{s}_t \mathrm{d}\boldsymbol{a}_t \rho(\boldsymbol{s}_0) \sum_{i=1}^n \omega_i(\boldsymbol{s}_t) \mathcal{N}(\boldsymbol{a}_t; \boldsymbol{\mu}_i(\boldsymbol{s}_t), \boldsymbol{\Sigma}_i(\boldsymbol{s}_t)) p(\boldsymbol{s}_{t+1}|\boldsymbol{s}_t, a_t) \sum_t \gamma^t r(\boldsymbol{s}_t, \boldsymbol{a}_t)$$

$$+ \beta \int \mathrm{d}\boldsymbol{a}_t \sum_{i=1}^n \omega_i(\boldsymbol{s}_t) \mathcal{N}(\boldsymbol{a}_t; \mu_i(\boldsymbol{s}_t), \Sigma_i(\boldsymbol{s}_t)) p(\boldsymbol{s}_{t+1}|\boldsymbol{s}_t, a_t)$$

$$\log \left( \sum_{i=1}^n \omega_i(s_t) \mathcal{N}(\boldsymbol{a}_t; \mu_i(\boldsymbol{s}_t), \Sigma_i(\boldsymbol{s}_t)) p(\boldsymbol{s}_{t+1}|\boldsymbol{s}_t, \boldsymbol{a}_t) \right). \qquad (25)$$

By inserting Eq. 25 into Eq. 24 we obtain for the individual parameters of the GMM

$$\nabla_{\boldsymbol{\mu}_l} J(\pi) = \mathbb{E}_\tau \left[ \sum_t \left( \frac{\omega_l \mathcal{N}(\boldsymbol{s}_t, \boldsymbol{a}_t; \boldsymbol{\mu}_l, \boldsymbol{\Sigma}_l) \boldsymbol{\Sigma}_l^{-1}((\boldsymbol{s}_t, \boldsymbol{a}_t) - \boldsymbol{\mu}_l)}{\sum_j \boldsymbol{\omega}_j \mathcal{N}(\boldsymbol{s}_t, \boldsymbol{a}_t; \boldsymbol{\mu}_j, \boldsymbol{\Sigma}_j)} \right. \right. \qquad (26)$$

$$\left. \left. - \frac{\omega_l \int \mathrm{d}\boldsymbol{a} \mathcal{N}(\boldsymbol{s}_t, \boldsymbol{a}_t; \boldsymbol{\mu}_l, \boldsymbol{\Sigma}_l) \boldsymbol{\Sigma}_l^{-1}((\boldsymbol{s}_t, \boldsymbol{a}_t) - \boldsymbol{\mu}_l)}{\sum_j \boldsymbol{\omega}_j \int \mathrm{d}\boldsymbol{a} \mathcal{N}(\boldsymbol{s}_t, \boldsymbol{a}_t; \boldsymbol{\mu}_j, \boldsymbol{\Sigma}_j)} \right) \sum_{t'>t} r(\boldsymbol{s}_t, \boldsymbol{a}_t) \right],$$

$$= \mathbb{E}_\tau \left[ \sum_t \left( \frac{\omega_l \mathcal{N}(\boldsymbol{s}_t, \boldsymbol{a}_t; \boldsymbol{\mu}_l, \boldsymbol{\Sigma}_l) \boldsymbol{\Sigma}_l^{-1}((\boldsymbol{s}_t, \boldsymbol{a}_t) - \boldsymbol{\mu}_l)}{\sum_j \boldsymbol{\omega}_j \mathcal{N}(\boldsymbol{s}_t, \boldsymbol{a}_t; \boldsymbol{\mu}_j, \boldsymbol{\Sigma}_j)} \right. \right. \qquad (27)$$

$$\left. \left. - \delta_{\boldsymbol{s}} \frac{\omega_l \mathcal{N}(\boldsymbol{s}_t; \boldsymbol{\mu}_{l,s} \boldsymbol{\Sigma}_{l,ss}) \boldsymbol{\Sigma}_{l,ss}^{-1}(\boldsymbol{s}_t - \boldsymbol{\mu}_{l,s})}{\sum_j \boldsymbol{\omega}_j \mathcal{N}(\boldsymbol{s}_t; \boldsymbol{\mu}_{j,s}, \boldsymbol{\Sigma}_{j,ss})} \right) \sum_{t'>t} r(\boldsymbol{s}_t, \boldsymbol{a}_t) \right],$$

$$= \mathbb{E}_\tau \left[ \sum_t \left( \zeta_{l,\boldsymbol{s}_t,\boldsymbol{a}_t} \boldsymbol{\Sigma}_l^{-1}((\boldsymbol{s}_t, \boldsymbol{a}_t) - \boldsymbol{\mu}_l) - \delta_s \zeta_{l,\boldsymbol{s}_t} \boldsymbol{\Sigma}_{l,ss}^{-1}(\boldsymbol{s}_t - \boldsymbol{\mu}_{l,s}) \right) \sum_{t'>t} r(\boldsymbol{s}_t, \boldsymbol{a}_t) \right].$$

Here $\delta_{\boldsymbol{s}} \in \{0, 1\}$ indicates which terms the gradient acts on. In this case, the gradient act on the state components and it is absent for the action dimensions.

$$\nabla_{\boldsymbol{\Sigma}_l} J(\pi) = \mathbb{E}_\tau \left[ \sum_t \left( -\frac{1}{2} \frac{\omega_l \mathcal{N}(\boldsymbol{s}_t, \boldsymbol{a}_t; \boldsymbol{\mu}_l, \boldsymbol{\Sigma}_l) \boldsymbol{\Sigma}_l^{-1} \left( 1 - ((\boldsymbol{s}_t, \boldsymbol{a}_t) - \boldsymbol{\mu}_l)((\boldsymbol{s}_t, \boldsymbol{a}_t) - \boldsymbol{\mu}_l)^\mathsf{T} \boldsymbol{\Sigma}_l^{-1} \right)}{\sum_j \boldsymbol{\omega}_j \mathcal{N}(\boldsymbol{s}_t, \boldsymbol{a}_t; \boldsymbol{\mu}_j, \boldsymbol{\Sigma}_j)} \right. \right.$$

$$(28)$$

$$\left. \left. + \frac{1}{2} \frac{\omega_l \int \mathrm{d}\boldsymbol{a} \mathcal{N}(\boldsymbol{s}_t, \boldsymbol{a}_t; \boldsymbol{\mu}_l, \boldsymbol{\Sigma}_l) \boldsymbol{\Sigma}_l^{-1} \left( 1 - ((\boldsymbol{s}_t, \boldsymbol{a}_t) - \boldsymbol{\mu}_l)((\boldsymbol{s}_t, \boldsymbol{a}_t) - \boldsymbol{\mu}_l)^\mathsf{T} \boldsymbol{\Sigma}_l^{-1} \right)}{\sum_j \boldsymbol{\omega}_j \int \mathrm{d}\boldsymbol{a} \mathcal{N}(\boldsymbol{s}_t, \boldsymbol{a}_t; \boldsymbol{\mu}_j, \boldsymbol{\Sigma}_j)} \right) \sum_{t'>t} r(\boldsymbol{s}_t, \boldsymbol{a}_t) \right],$$

$$= \mathbb{E}_\tau \left[ \sum_t \left( -\frac{\zeta_{l,\boldsymbol{s}_t,\boldsymbol{a}_t}}{2} \boldsymbol{\Sigma}_l^{-1} \left( 1 - ((\boldsymbol{s}_t, \boldsymbol{a}_t) - \boldsymbol{\mu}_l)((\boldsymbol{s}_t, \boldsymbol{a}_t) - \boldsymbol{\mu}_l)^\mathsf{T} \boldsymbol{\Sigma}_l^{-1} \right) \right. \right.$$

$$\left. \left. + \delta_s \frac{\zeta_{l,\boldsymbol{s}_t}}{2} \boldsymbol{\Sigma}_{l,s}^{-1} \left( 1 - (\boldsymbol{s}_t - \boldsymbol{\mu}_{l,s})(\boldsymbol{s}_t - \boldsymbol{\mu}_{l,s})^\mathsf{T} \boldsymbol{\Sigma}_{l,s}^{-1} \right) \right) \sum_{t'>t} r(\boldsymbol{s}_t, \boldsymbol{a}_t) \right].$$

$$\nabla \boldsymbol{\omega}_l J(\pi) = \mathbb{E}_\tau \left[ \sum_t \left( \frac{\mathcal{N}(\boldsymbol{s}_t, \boldsymbol{a}_t; \boldsymbol{\mu}_l, \boldsymbol{\Sigma}_l)}{\sum_j \boldsymbol{\omega}_j \mathcal{N}(\boldsymbol{s}_t, \boldsymbol{a}_t; \boldsymbol{\mu}_j, \boldsymbol{\Sigma}_j)} - \frac{\int d\boldsymbol{a} \mathcal{N}(\boldsymbol{s}_t, \boldsymbol{a}_t; \boldsymbol{\mu}_l, \boldsymbol{\Sigma}_l)}{\sum_j \boldsymbol{\omega}_j \int d\boldsymbol{a} \mathcal{N}(\boldsymbol{s}_t, \boldsymbol{a}_t; \boldsymbol{\mu}_j, \boldsymbol{\Sigma}_j)} \right) \sum_{t'>t} r(\boldsymbol{s}_t, \boldsymbol{a}_t) \right]$$
(29)

$$= \mathbb{E}_\tau \left[ \sum_t \frac{(\zeta_{l,\boldsymbol{s}_t,\boldsymbol{a}_t} - \zeta_{l,\boldsymbol{s}_t})}{\boldsymbol{\omega}_l} \sum_{t'>t} r(\boldsymbol{s}_t, \boldsymbol{a}_t) \right].$$
(30)

Note that we introduced the responsibilities $\zeta_{l,\boldsymbol{s}_t,\boldsymbol{a}_t}$ and $\zeta_{l,\boldsymbol{s}_t}$, which are defined as follows

$$\zeta_{l,\boldsymbol{s}_t,\boldsymbol{a}_t} = \frac{\boldsymbol{\omega}_l \mathcal{N}(\boldsymbol{s}_t, \boldsymbol{a}_t; \boldsymbol{\mu}_l, \boldsymbol{\Sigma}_l)}{\sum_j \boldsymbol{\omega}_j \mathcal{N}(\boldsymbol{s}_t, \boldsymbol{a}_t; \boldsymbol{\mu}_j, \boldsymbol{\Sigma}_j)}, \quad \text{and}$$
(31)

$$\zeta_{l,\boldsymbol{s}_t} = \frac{\boldsymbol{\omega}_l \int d\boldsymbol{a} \mathcal{N}(\boldsymbol{s}_t, \boldsymbol{a}_t; \boldsymbol{\mu}_l, \boldsymbol{\Sigma}_l)}{\sum_j \boldsymbol{\omega}_j \int d\boldsymbol{a} \mathcal{N}(\boldsymbol{s}_t, \boldsymbol{a}_t; \boldsymbol{\mu}_j, \boldsymbol{\Sigma}_j)} = \frac{\boldsymbol{\omega}_l \mathcal{N}(\boldsymbol{s}_t; \boldsymbol{\mu}_{l,s}, \boldsymbol{\Sigma}_{l,ss})}{\sum_j \boldsymbol{\omega}_j \mathcal{N}(\boldsymbol{s}_t; \boldsymbol{\mu}_{j,s}, \boldsymbol{\Sigma}_{j,ss})}.$$
(32)

## A.4 RELATION BETWEEN FORWARD AND BACKWARD DISCRETIZATION IN THE BURES-WASSERSTEIN METRIC

In this section we outline the relation between the implicit and explicit optimization schema w.r.t. the Bures-Wasserstein metric, which is leveraged to formulate our policy optimization in § 3. We closely follow Chen & Li (2020). For the sake of simplicity, we group the Gaussian parameters $\boldsymbol{\mu}$ and $\boldsymbol{\Sigma}$ into a single parameter vector $\boldsymbol{\theta}$. Furthermore, we restrict our explanation to a single Gaussian component, which is possible without loosing generality, as each of the $N$ components live in its own manifold $\mathbb{R}^d \times \mathcal{S}^d_{++}$. The Riemannian gradient w.r.t the Gaussian parameters $\boldsymbol{\theta}$, $\text{grad}_{\boldsymbol{\theta}} J(\pi(\boldsymbol{\theta}))$, satisfies by definition

$$g_{\boldsymbol{\theta}}(\text{grad}_{\boldsymbol{\theta}} J(\pi(\boldsymbol{\theta}), \boldsymbol{\xi}) = \nabla_{\boldsymbol{\theta}} J(\pi(\boldsymbol{\theta})) \cdot \boldsymbol{\xi},$$
(33)

where $\nabla_{\boldsymbol{\theta}}$ denotes the Euclidean gradient, $\boldsymbol{\xi}$ is an arbitrary vector on the tangent space $\mathcal{T}_{\boldsymbol{\theta}}\mathcal{M}$, and $g_{\boldsymbol{\theta}}$ is the Riemannian metric tensor, defining the inner product on $\mathcal{T}_{\boldsymbol{\theta}}\mathcal{M}$. The Riemannian metric $g_{\boldsymbol{\theta}}$ can be written as

$$g_{\boldsymbol{\theta}}(\boldsymbol{\zeta}, \boldsymbol{\xi}) = \boldsymbol{\zeta}^\top G_W(\boldsymbol{\theta}) \boldsymbol{\xi},$$
(34)

with two arbitrary tangent vectors $\boldsymbol{\zeta}, \boldsymbol{\xi}$, and $G_W(\boldsymbol{\theta})$ being a positive definite matrix. Moreover, note that the Wasserstein distance $W_2^2(\mathcal{N}(\boldsymbol{\theta}), \mathcal{N}(\boldsymbol{\theta} + \Delta\boldsymbol{\theta}))$, where $\Delta\boldsymbol{\theta}$ denotes a small perturbation in the Gaussian parameters $\boldsymbol{\theta}$, can be expressed as

$$W_2^2(\mathcal{N}(\boldsymbol{\theta}), \mathcal{N}(\boldsymbol{\theta} + \Delta\boldsymbol{\theta})) = \frac{1}{2}(\Delta\boldsymbol{\theta}^\top) G_W(\boldsymbol{\theta})(\Delta\boldsymbol{\theta}) + O((\Delta\boldsymbol{\theta})^2),$$
(35)

for $\Delta\boldsymbol{\theta} \to 0$. Similarly, we can approximate the objective evaluated at $J(\boldsymbol{\theta} + \Delta\boldsymbol{\theta})$ via the Taylor theorem as

$$J(\boldsymbol{\theta} + \Delta\boldsymbol{\theta}) = J(\boldsymbol{\theta}) + \nabla_{\boldsymbol{\theta}} J(\boldsymbol{\theta}) \cdot \Delta\boldsymbol{\theta} + O((\Delta\boldsymbol{\theta})^2).$$
(36)

With this, we can approximate

$$\boldsymbol{\theta}_{k+1} = \underset{\boldsymbol{\theta}}{\arg\min} \left( \frac{W_2^2(\pi(\boldsymbol{\theta}), \pi(\boldsymbol{\theta}_k))}{2\tau} - J(\pi(\boldsymbol{\theta})) \right),$$
(37)

$$\approx \underset{\boldsymbol{\theta}}{\arg\min} \left( \frac{(\boldsymbol{\theta} - \boldsymbol{\theta}_k)^\top G_W (\boldsymbol{\theta} - \boldsymbol{\theta}_k)}{2\tau} - \nabla_{\boldsymbol{\theta}} J(\boldsymbol{\theta}) \cdot (\boldsymbol{\theta} - \boldsymbol{\theta}_k) \right),$$
(38)

from which we obtain the update equation for $\boldsymbol{\theta}$ as follows

$$\boldsymbol{\theta}_{k+1} = \boldsymbol{\theta}_k + \tau G_W(\boldsymbol{\theta}_k)^{-1} \nabla_{\boldsymbol{\theta}} J(\pi(\boldsymbol{\theta})).$$
(39)

Note that Eq. 39 in turn corresponds to an approximation of the exact Riemannian gradient descent

$$\boldsymbol{\theta}_{k+1} = R_{\boldsymbol{\theta}_k} (\lambda \cdot \text{grad}_{\boldsymbol{\theta}} J(\pi(\boldsymbol{\theta}_k))).$$
(40)

This approximation can be obtained by considering a first-order approximation of the geodesic on the $BW$ manifold. As the exponential map (a.k.a. the retraction) is defined via the geodesic, the retraction operator in Eq. 40 turns into a simple addition operation under a first-order approximation, leading to Eq. 39. Notice that such approximation does not guarantee that the updated parameters $\boldsymbol{\theta}$ stay on the manifold, except for the cases in which $\boldsymbol{\theta} \in \mathbb{R}^d$. In our case, we leverage the retraction and Riemannian gradients of (Malagò et al., 2018; Han et al., 2021), which allow us to apply the exact Riemannian gradient descent of 40. This avoids to rely on first-order approximations and in turn we can guarantee that the updates of the Gaussian distribution parameters always lie on on the product manifold $\left(\mathbb{R}^d \times \mathcal{S}_{++}^d\right)^N$.

### A.5 ADDITIONAL DETAILS ON THE IMPLEMENTATION

We extended the Pymanopt (Townsend et al., 2016) by adding a custom line-search routine that accounts for a constraint on the Wasserstein distance between the old and the optimized GMMs. The details of this line-search can be found in Algorithm 2.

---

**Algorithm 2** Constrained line-search. The constraint function $c(x_0, \cdot)$ is arbitrary in general. We use the $L^2$-Wasserstein distance between two points on the manifold of GMMs as constraint.

    **Input**: point $\boldsymbol{x_0}$ on the manifold, descent direction $\boldsymbol{d}$, initial step size $\lambda_0$, decrement $\alpha$, constraint $c(\boldsymbol{x_0}, \cdot)$, maximum allowed value for constraint $c_{\max}$, minimum step size $\lambda_{\min}$
    **Output**: step size $s$, updated point on manifold $\boldsymbol{x}$

1:  $\boldsymbol{x} = \boldsymbol{x_0} + \lambda_0 \cdot \boldsymbol{d}$
    $\lambda = \lambda_0$
2:  **while** $c(\boldsymbol{x_0}, \boldsymbol{x}) > c_{\max}$ and $\lambda > \lambda_{\min}$ **do**
3:     decrease step size: $\lambda = \alpha \cdot \lambda$
       update point on manifold: $\boldsymbol{x} = \boldsymbol{x_0} + \lambda \cdot \boldsymbol{d}$
4:  **end while**
5:  **if** $\lambda < \lambda_{\min}$ **then**
6:     return $\lambda_0, \boldsymbol{x_0}$
7:  **else**
8:     return $\lambda, \boldsymbol{x}$
9:  **end if**

---

### A.6 ADDITIONAL DETAILS ON EXPERIMENTS

#### A.6.1 ADDITIONAL RESULTS

Fig. 5 shows the convergence curves for the two baselines as in Fig. 3 of the main paper, however, we extended the horizontal axis up to the maximum number of environment steps used for training.

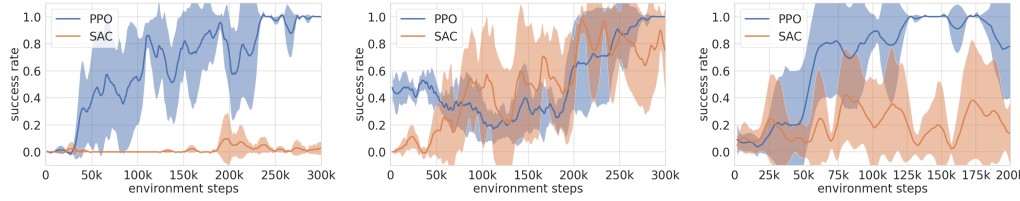

Figure 5: The success rate of the two baselines on the reaching task (*left*), the collision-avoidance task (*middle*) and the multiple-goal task (*right*). The shaded area indicates the standard deviation over 5 runs.

Fig. 6 shows the variance of the success rate for the three methods at their time step of convergence for all three robotic tasks. Concerning SAC, which did not converge after the maximum number of environment steps used for training, we chose the last time step. Specifically, we chose the following time steps for PPO, SAC and WGF, respectively: reaching task $(280000, 400000, 80000)$, collision avoidance task $(275000, 300000, 90000)$, multiple goal task $(130000, 200000, 95000)$. These plots show that PPO may also reach low-variance success rate over the five runs at the time step of conver-

gence, at the cost of a prohibitively large number of steps. SAC showed huge variance in all tasks, apart from the reaching task, where all runs collapsed to a success rate of 0.

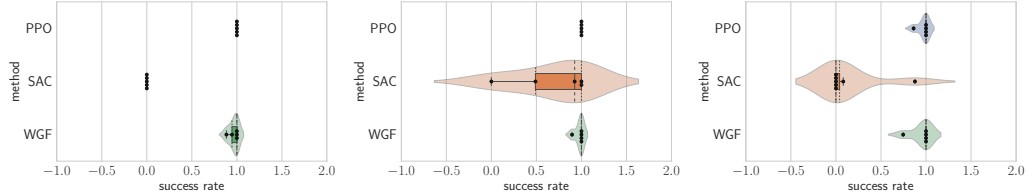

Figure 6: Variance of the success rate over the 5 runs for our method (WGF) and the two baselines on the reaching task (*left*), the collision avoidance task (*middle*) and the multiple-goal task (*right*). The violine plots are overlaid with box plots, quartile lines and a swarm plot, where dots indicate the success rates of individual runs. The time steps at which we determined the variance are for PPO, SAC and WGF for the three tasks from left to right: $(280000, 400000, 80000)$, $(275000, 300000, 90000)$, $(130000, 200000, 95000)$.

### A.6.2 ADDITIONAL ABLATIONS

In order to assess the influence of leveraging a Riemannian optimization approach on the Bures-Wasserstein manifold, we conducted an ablation of our method by eliminating the Riemannian formulation. Instead of the explicit Euler scheme update in Eq. 12, which corresponds to Riemannian gradient descent w.r.t. the Bures-Wasserstein metric, we use the implicit Euler scheme

$$\hat{\boldsymbol{\mu}}_{k+1} = \arg\min_{\hat{\boldsymbol{\mu}}} \left( \frac{W_2^2(\pi_k(\hat{\boldsymbol{\mu}}), \pi_k)}{2\tau} - J(\pi_k(\hat{\boldsymbol{\mu}})) \right), \tag{41}$$

$$\hat{\boldsymbol{\Sigma}}_{k+1} = \arg\min_{\hat{\boldsymbol{\Sigma}}} \left( \frac{W_2^2(\pi_k(\hat{\boldsymbol{\Sigma}}), \pi_k)}{2\tau} - J(\pi_k(\hat{\boldsymbol{\Sigma}})) \right). \tag{42}$$

To guarantee that the updated covariance matrices do not leave the manifold of symmetric positive definite matrices, we parameterize them in terms of Cholesky factors. The results obtained with this non-Riemannian version of our method are shown in Fig. 7 in direct comparison to our method and Fig. 8 for an extended range.

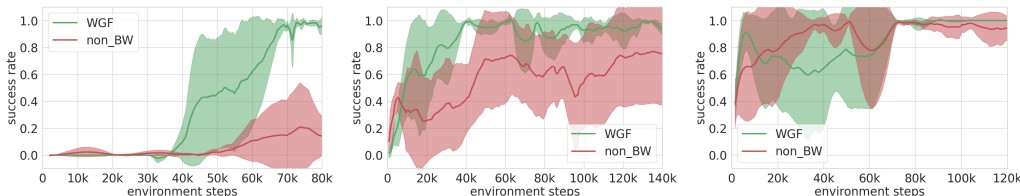

Figure 7: The success rate of our method and an ablated version, not using the Bures-Wasserstein formulation for the reaching task (*left*), the collision-avoidance task (*middle*) and the multiple-goal task (*right*). The shaded area indicates the standard deviation over 5 runs.

The results clearly show that the non-Riemannian method struggles to reach a success rate of 1 for the reaching task and the collision-avoidance task. Furthermore, we observe a high variance over different runs in the same settings (see Fig. 9 and Fig. 10). We attribute this to the fact that the our method takes exact gradient steps in the direction of steepest descent w.r.t. the underlying BW metric, whereas the implicit scheme only approximates this direction. For this reason the non-Riemannian method is much more noisy, which in turn leads to the aforementioned high variance. Nevertheless, the multiple-goal task constitutes an exception. Here we observed a similar performance for our approach and the ablated method. The reason for this is that the optimization of this task is mainly dominated by the weight updates, which are identical for both methods. This result is therefore expected and confirms that correctness of our ablation strategy.

### A.6.3 ADDITIONAL EXPERIMENT WITH 7-DoF ROBOTIC MANIPULATOR

We carried out an additional experiment to show that our method can be employed on tasks performed by off-the-shelf robotic manipulators (e.g. a 7-DoF Franka Emika Panda robot). Specifi-

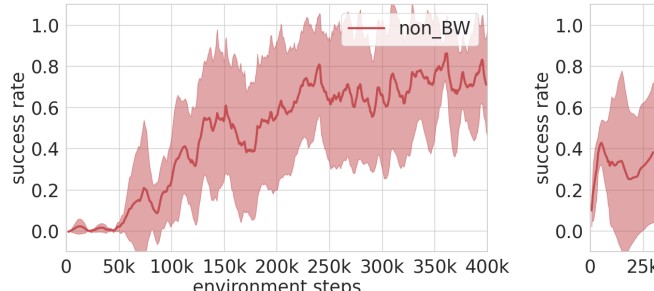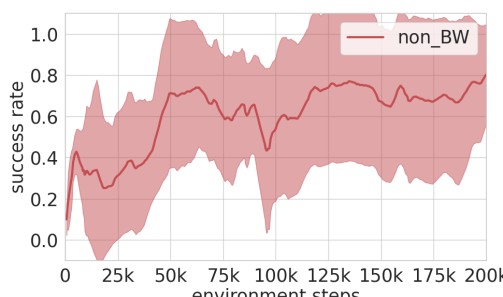

Figure 8: Extended plot of the success rate of and ablated version of our method, not using the Bures-Wasserstein-based formulation for the reaching task (*left*), the collision-avoidance task (*middle*) and the multiple-goal task (*right*). The shaded area indicates the standard deviation over 5 runs.

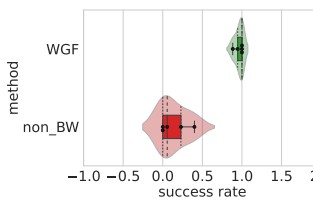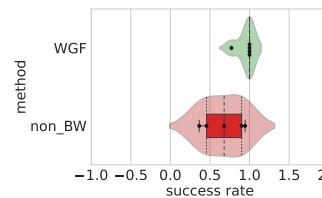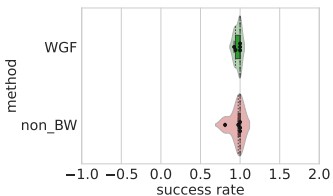

Figure 9: Variance of the success rate over 5 runs for our method (WGF) and the ablated method (non-BW) on the reaching task (*left*), the collision avoidance task (*middle*) and the multiple-goal task (*right*). The violine plots are overlaid with box plots, quartile lines and a swarm plot, where dots indicate the success rates of individual runs. The time steps at which we determined the variance are 80000, 90000, 85000.

cally, we extended the collision-avoidance task described in § 4 to a 3D environment (i.e. the state $s = x \in \mathbb{R}^3$ and the action $a = \dot{x} \in \mathbb{R}^3$). The initial 3-components GMM policy was trained using 10 human demonstrations featuring linear reaching 3D trajectories. For policy optimization, we used a sparse reward defined as a function of the position error between the robot end-effector position and the target at the end of the rollout. Moreover, two sparse penalty terms were added to punish collision with obstacles and divergent trajectories.

Similarly to the planar task reported in the main paper, we tested whether our method was able to adapt a trajectory tracking skill in order to avoid collisions with newly added obstacles. This means that the robot end-effector needed to pass through a narrow path between two spherical obstacles. The robot end-effector pose was controlled using a full-pose Cartesian velocity controller at a frequency of 100Hz, where the end-effector orientation was kept constant. Figure 11 shows that our method reached a success rate of 1.0 very quickly, taking approximately 20000 environment steps. Moreover, the solution variance of our method was also very low, which is consistent with our observations concerning the performance of our policy optimization on the three planar tasks analyzed in the main paper.

### A.6.4 INITIAL GMM POLICIES

For the sake of completeness, Fig. 12 provides 2D projections of the initial GMM policies learned from demonstrations for the three robotic settings considered in the main paper: the reaching motion skill, the collision-free trajectory tracking, and the multiple-goal task. Figure 12 also provides the demonstration data used to train the initial policies. Note that these models are then adapted according to the policy optimization approach introduced in § 3.2.

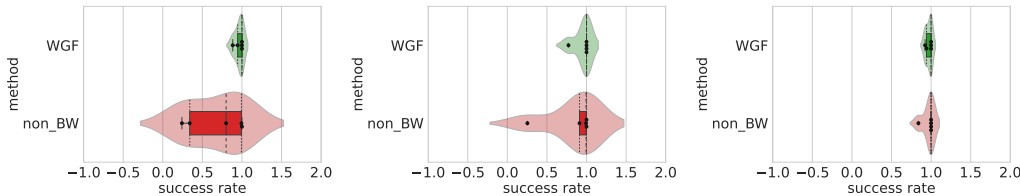

Figure 10: Variance of the success rate over 5 runs for our method (WGF) and the ablated method (non-BW) on the reaching task (*left*), the collision avoidance task (*middle*) and the multiple-goal task (*right*). The violine plots are overlaid with box plots, quartile lines and a swarm plot, where dots indicate the success rates of individual runs. The time steps at which we determined the variance are $(80000, 400000)$, $(90000, 200000)$, $(85000, 90000)$.

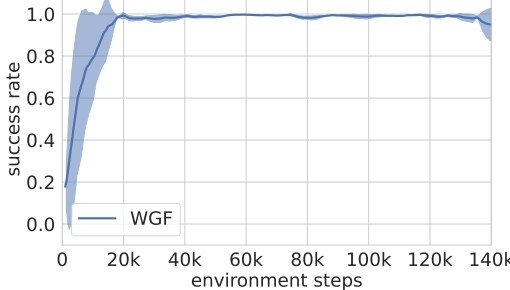

Figure 11: The success rate of our method applied to the 3D narrow-path task performed by the 7-DoF Panda robotic manipulator. The shaded area indicates the standard deviation over 5 runs.

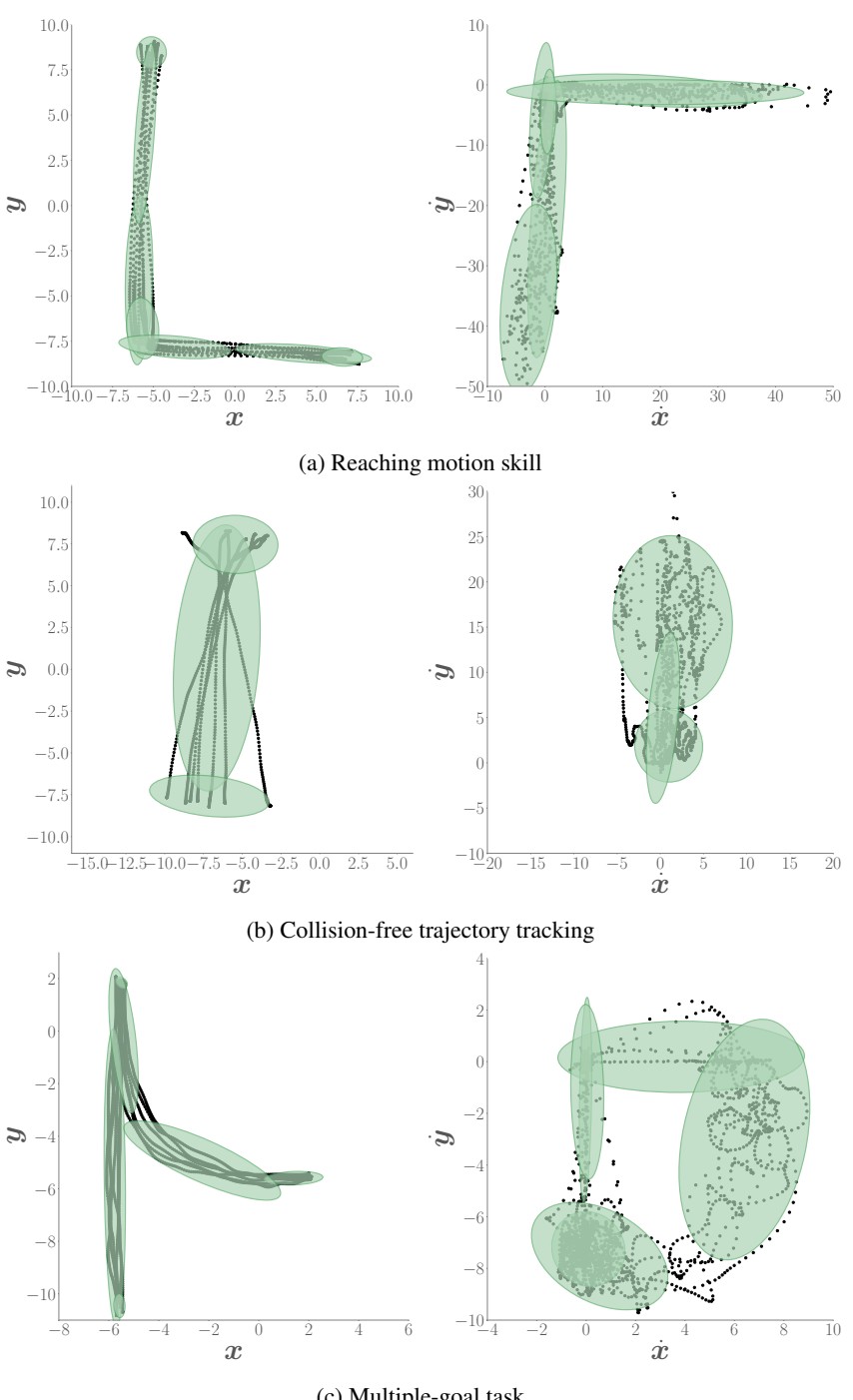

(a) Reaching motion skill

(b) Collision-free trajectory tracking

(c) Multiple-goal task

Figure 12: Green Gaussian components ( ) represent the initial GMM policy learned from demonstrations, projected on the Cartesian position (*left*) and velocity (*left*) spaces. The recorded position and velocity data are depicted as black dots (.).

