# OpenReview forum: "Wasserstein Gradient Flows for Optimizing GMM-based Policies"
_ICLR.cc/2023/Conference — Submitted to ICLR 2023_

### Official Review · Reviewer_1kyr · 2022-10-24

**Confidence:** 4
**Correctness:** 3
**Technical Novelty And Significance:** 3
**Empirical Novelty And Significance:** 3
**Recommendation:** 3

**Clarity, Quality, Novelty And Reproducibility:**

Updated: Overall paper is well-written. The approach is novel for optimizing GMM-based robot policies. However, the experiments need significant improvement to validate the proposed approach under complex collision avoidance constraints.



**Strength And Weaknesses:**

*Strength*
+ A novel Gradient flow approach for GMM optimization
+ Algothrim seems sound with theoretical backing.
+ The results demonstrate better performance than PPO and SAC

*Weaknesses*
- Experiment section is relatively weak. The robot tasks are too simple. Although WGF outperforms PPO and SAC on those tasks, a more cluttered scenario with a realistic robot arm (URDFs of UR5, Panda, etc.) would exhibit the applicability of the proposed approach to practical environments. Perhaps a relevant baseline for simulation environment setup could be [1].

[1] Continuous-time Gaussian process motion planning via probabilistic inference. The International Journal of Robotics Research, 37(11), 1319-1340.

**Summary Of The Paper:**

This paper presents GMM-based robot policy optimization formulated as a Wasserstein gradient flow, resulting in constraining policy updates for a stable optimization process. The proposed method is compared against two baselines PPO-based GMM update and SAC-GMM. The evaluations are performed in three tasks with a toy robot arm to reach a target, avoid collision during trajectory tracking, and reach multiple targets. The results demonstrate that their proposed approach (WGF) outperforms PPO and SAC in all three tasks.

**Summary Of The Review:**

This paper presents a stable approach to optimize GMM policies for robot control which also outperforms prior methods such as PPO and SAC. However, the experiments could have more complex environments to highlight the proposed approach's scalability better. Even if it does not scale, those evaluations will highlight a limitation to address in future work.

Updated: The paper presents an interesting idea but lacks experiments that could validate the proposed approach under standard collision avoidance constraints for motion planning problems.

---

### Official Review · Reviewer_zFF5 · 2022-10-28

**Confidence:** 3
**Clarity, Quality, Novelty And Reproducibility:** I think the method is novel.
**Correctness:** 3
**Technical Novelty And Significance:** 3
**Empirical Novelty And Significance:** 3
**Recommendation:** 5

**Strength And Weaknesses:**

Strength:

- This paper attacks a very important direction, optimizing a policy with hidden structures. Though the paper only studied Gaussian policy here, I can imagine that a similar idea can be applied to other policy classes, like a neural network.
- The EM-like optimization approach for optimizing GMM with REINFORCE is novel to me. I think this might be the key to its success, as directly optimizing the likelihood of GMM models are not easy.
- Please correct me if I am wrong. The proposed Riemannian ensures the optimization does not break the semidefinite constraints of Gaussian models. Optimizing GMM parameters in the BW manifolds constrain the policy update, enabling better transfer performance.

Weakness:

- The GMM policy can hardly generalize to high-dimensional tasks, which will constrain its application. It will be interesting if it is possible to optimize broader policies.
- I feel the paper lacks the necessary ablation study. The baseline comparison is reasonable. However, it is unclear why the proposed method works by comparing it with SAC. I do not fully understand why the Reinmannian gradient is necessary. Is it possible to optimize the policy by parameterizing the Sigma as a form of V^TV to guarantee its positive semidefinite? What would happen? Besides, what would happen if the Wasserstein part is removed from eq (13)? An ablation study is required to illustrate the contribution of each component.

**Summary Of The Paper:**

This paper proposes a Riemannian optimization method to optimize GMM policies. It uses GMM to represent the policy structure and demonstrates an EM-like approach to optimize such a policy with hidden variables with a maximum entropy RL objective. In each iteration, the Gaussian parameters are optimized with the Riemannian gradients, while the GMM parameters are optimized later with an additional Wasserstein distance constraint. Optimizing GMM with RL objectives enables the authors to transfer a GMM policy from demonstrations to new tasks. The authors considered several reach or collision avoidance tasks and showed better performance.

**Summary Of The Review:**

The paper proposes an interesting idea of how to optimize a policy with prior structures. However, I feel the paper lacks enough ablation study and can not be accepted given its current form. I am happy to increase the score if additional results are provided.

---

### Official Review · Reviewer_8XBw · 2022-11-05

**Confidence:** 4
**Correctness:** 3
**Technical Novelty And Significance:** 2
**Empirical Novelty And Significance:** 4
**Recommendation:** 5

**Clarity, Quality, Novelty And Reproducibility:**

The paper is well-written and clear. It would have been beneficial if there was a clearer introduction to the nomenclature associated with policy optimization and RL for the non-expert reader.

Bures-Wasserstein gradient flows and embedding Gaussian mixtures on the Bures-Wasserstein manifold have been studied extensively. However, making these connections in the context of RL and policy optimization are novel and certainly useful.

**Strength And Weaknesses:**

**Strengths:**
* The paper is well-written and easy to follow. I appreciate the informative, yet concise, introduction on Wassserstein gradient flows for the non-expert reader.

* Even though the idea of formulating policy optimization using Wasserstein gradient flows is already known, the use of the Bures-Wasserstein geometry in the specific setting of Gaussian mixture models is novel and interesting.

* Despite the drawbacks in the experiments (see comments below) the authors do a nice job interpreting the results and providing useful insights.

* The authors claim that their perspective leads to significant computational advantage. (This would have been more compelling if it were reported as a benchmark of time taken for each method).


**Weaknesses:**
* My biggest concern is on the theoretical front. The objective of the paper is to study Wasserstein gradient flows which don't provide any dynamics for the weight update. A weight update, as in Eqs. (13 & 14), will warrant machinery from Wasserstein Fisher-Rao gradient flows instead. See, for example,  [Chizat et al. (2018)](https://link.springer.com/article/10.1007/s10208-016-9331-y) [Liero et al. (2018)](https://link.springer.com/article/10.1007/s00222-017-0759-8) [Chizat (2022)](https://link.springer.com/article/10.1007/s10107-021-01636-z) Therefore, it is unsurprising that the authors note, in their conclusions section, that the "weight update often hampered with the update of the Gaussian parameters".

* Notwithstanding, for the proposed method using the Bures-Wasserstein geometry, I would really have liked to see some guarantee that the proposed methodology leads to policy improvement, c.f., Wang et al. (2020, Theorem 2).

* On the experimental front, I found the benchmarks to be lacking. For example, it would have been instructive if the authors compared their method to other parametric policy updates such as [Moskovitz (2020)](https://arxiv.org/abs/2010.05380) ; especially since the notion of Wasserstein natural gradients (Chen & Li, 2020) is mentioned in §3.1 and §A.4.

* Furthermore, it would have been even more enlightening if the authors contrasted their performance to that of Wang et al. (2020), since this would have demonstrated the advantage of the Bures-Wasserstein perspective in contrast to their perspective which relies on the JKO scheme.



**Minor Clarifications:**
* I'm unclear on how the number of Gaussian components, $K$, are chosen for the mixture model. For example, Delon & Desolneux (2020, §6.3) note that performance is sensitive to the choice of $K$. Alternatively, Figueroa & Billard (2018) tackle this issue by choosing a Dirichlet process prior on $K$. How sensitive are the results to this choice?

* The authors implement the JKO scheme for the weight update using the Sinkhorn algorithm, which includes an entropic regularization. How does this interplay with the entropic regularization in the free energy functional J? The proximal map is effectively subtracting  a factor $\beta$  of the entropy term due to $J_{r, \beta}(\pi)$ and adding a $1/2\tau$ factor due to Sinkhorn?


**Summary Of The Paper:**

The authors consider the problem of policy optimization; they propose using Bures-Wasserstein gradient flows for optimizing policies which are constrained to the class of Gaussian mixture models. Existing methods in this setting don't exploit the structure enforced by the Gaussian mixture models, and the authors demonstrate the advantage of this perspective through empirical analyses on 2$d$ benchmarks.

**Summary Of The Review:**

The authors propose an interesting method for policy optimization through the lens of Bures-Wasserstein gradient flows. While the theoretical underpinnings are unclear, the experiments show useful improvement over _some_ benchmarks. It would have been even more compelling if the experiments were more exhaustive.

---

### Decision · Program_Chairs · 2023-01-20

**Decision:**

Reject

**Justification For Why Not Higher Score:**

The paper needs improvement both on theory end and experimentation end.

**Justification For Why Not Lower Score:**

N/A

**Metareview: Summary, Strengths And Weaknesses:**

The paper proposes to use Wasserstein gradient flows for learning policies constrained to the class of mixture of gaussian distribution. Previous work don't consider the geometric structure of the policy class and authors propose to use Bures Wasserstein geometry and optimize via a Riemannian optimization.

**Strength of the work:**

Using the geometry of the policy in RL using gradient flow is an interesting direction. The paper is easy to follow.

**Weakness**

There was multiple concerns regarding the work:

1- The weights updates that is decoupled from the geometry of gaussian and for Wasserstein gradient Flow. As pointed by Reviewer 8XBw , Wasserstein Fisher Rao geometry is more adapted for this. A comparison of the Bur

2-  More comparison to Wang et al 2020 as well as more benchmarking and ablation of the method

3- Applications in robotics were not convincing as pointed by reviewer 1kyr,

**Summary Of Ac-Reviewer Meeting:**

N/A